# Learning-Based Seismic Velocity Inversion with Synthetic and Field Data

**DOI:** 10.3390/s23198277

**Published:** 2023-10-06

**Authors:** Stuart Farris, Robert Clapp, Mauricio Araya-Polo

**Affiliations:** 1Department of Geophysics, Stanford University, Stanford, CA 94305, USA; bob@sep.stanford.edu; 2TotalEnergies, EP R&T, Houston, TX 77002, USA; mauricio.araya@totalenergies.com

**Keywords:** field data, deep learning, synthetic training data, seismic propagation velocity, inverse problems

## Abstract

Building accurate acoustic subsurface velocity models is essential for successful industrial exploration projects. Traditional inversion methods from field-recorded seismograms struggle in regions with complex geology. While deep learning (DL) presents a promising alternative, its robustness using field data in these complicated regions has not been sufficiently explored. In this study, we present a thorough analysis of DL’s capability to harness labeled seismograms, whether field-recorded or synthetically generated, for accurate velocity model recovery in a challenging region of the Gulf of Mexico. Our evaluation centers on the impact of training data selection and data augmentation techniques on the DL model’s ability to recover velocity profiles. Models trained on field data produced superior results to data obtained using quantitative metrics like Mean Squared Error (MSE), Structural Similarity Index Measure (SSIM), and $R^{2}$ (R-squared). They also yielded more geologically plausible predictions and sharper geophysical migration images. Conversely, models trained on synthetic data, while less precise, highlighted the potential utility of synthetic training data, especially when labeled field data are scarce. Our work shows that the efficacy of synthetic data-driven models largely depends on bridging the domain gap between training and test data through the use of advanced wave equation solvers and geologic priors. Our results underscore DL’s potential to advance velocity model-building workflows in industrial settings using previously labeled field-recorded seismograms. They also highlight the indispensable role of earth scientists’ domain expertise in curating synthetic data when field data are lacking.

## 1. Introduction

Building an accurate velocity model of the subsurface is a critical step for many applications such as geophysical exploration and monitoring of geologically stored CO${}_{2}$, having a direct impact on the quality of seismic images and subsequent geological interpretation [1,2,3]. Conventional velocity model-building methods, such as normal moveout correction, tomographic inversion, and full waveform inversion, have served the aforementioned applications well in various settings [4,5,6,7,8]. However, these traditional techniques are largely based on linearized inversion and simplifying assumptions, leading to models that may inadequately capture the full range of subsurface complexities, especially in regions with complex geologic features, including salt diapers, basalt formations, and karst systems [5,6,9]. This limitation can impede seismic imaging and, subsequently, the understanding of valuable sedimentary reservoirs overlain by such complex structures.

Applying learning-based approaches to velocity model-building offers an intriguing solution to this challenge. Deep learning (DL), a sub-field of machine learning, uses multi-layered neural networks to learn and model complex, nonlinear relationships between input data and target outputs [10,11]. Multiple fields have leveraged DL to solve difficult inverse problems [12,13], including medical imaging [14,15], computational photography [16,17], and geophysics [18,19,20]. Solving inverse problems with DL begins with preparing a labeled training dataset consisting of pairs of input features and corresponding output labels. When real-world data are scarce because of cost and privacy concerns or are intractable to label, they can be generated synthetically [21,22,23].

Inverting for velocity models from seismic data using DL is an active area of research, but has only shown results on synthetic data [24,25,26,27,28,29,30] or field data in simple geologic regimes [31,32,33,34].

### Novel Contributions

The primary contribution of this paper lies in the innovative application of DL to velocity model inversion from field-recorded seismic data in a region with complex geological overburden in the U.S. Gulf of Mexico. Building upon the aforementioned research in learning-based inversion methods, this work extends the application of these techniques from synthetic to real-world seismic data, demonstrating that learning-based inversion can be applied to industrial-sized problems in complex geologic regimes.

By successfully applying DL models to real-world seismic data, we demonstrate the potential of DL to contribute to cost-effective and efficient imaging of valuable reservoirs, particularly those overlain by complex geology. Such reservoirs are not only of interest for hydrocarbon exploration, but also play a crucial role in permanent carbon sequestration [35,36], hydrogen gas storage [37,38], and geothermal prospecting [39,40,41].

In addition to applying DL to field data, this research uniquely presents a comprehensive performance comparison between models trained on synthetic data and those trained on field data. Our work highlights the advantages and challenges of each approach, illustrating that, while models trained with field data deliver superior performance, synthetic data remain a valuable alternative. In agreement with results from previous studies, we show that this is especially true when synthetic training data are generated with a high level of sophistication to carefully match the embedded features of field data [33,42]. The ability to effectively use synthetic data as a training substitute opens up possibilities for organizations with limited access to field data, while also emphasizing the need to closely resemble real-world physics in the generation of synthetic data.

Additionally, this work showcases the potential to repurpose previous work on building seismic velocity models as labeled data for training machine learning models. The demonstrated application underscores the ability of learning-based inversion methods to address industrial-sized problems, potentially revolutionizing the cost dynamics associated with imaging subsurface reservoirs.

## 2. Materials and Methods

### 2.1. Field Data and Legacy Model

We use a seismic dataset, made open-source by BP, from the Tiber field in the deepwaters of the U.S. Gulf of Mexico as the focus our study due to the region’s geologically complex overburden [43]. The 3D seismic dataset includes a legacy seismic velocity model that only partially overlaps with the corresponding seismic data, as visualized in Figure 1a,b. The seismic data collected from the Tiber field have been preprocessed following standard industry practices, including denoising, deghosting, direct arrival removal, and 5D interpolation. A representative shot gather is illustrated in Figure 1c, with energy from 3–60 Hz. We spatially divide this dataset into training and testing folds to allow for robust model validation. Though incomplete and partially accurate, the legacy model in the test fold is used to validate the results of the learning-based inversion approach.

### 2.2. Learning-Based Inversion Approach

Consider a seismic field experiment, f, applied to a velocity model, $x\in R^{n}$, resulting in measured seismic signals, $y\in R^{m}$:(1)$$\begin{array}{c} y=f(x)+\epsilon \end{array}$$

The goal of seismic velocity inversion is to recover an estimate of x from y. Solving this inverse problem is challenging for a variety of reasons. If m<n, the problem may be under-determined, implying that there are not enough observations to uniquely determine the velocity model. Additionally, the solution can be highly sensitive to noise, represented by ϵ, especially in the context of ill-posed problems. It might also be challenging to (or unknown how to) accurately model the operator f. Furthermore, modeling f can be computationally expensive, especially for large-scale seismic experiments. One approach to solving the inverse problem involves minimizing $L(x,g_{\theta }\left(y\right))$ for a suitable $g_{\theta }\left(\cdot \right)$, which assumes the role of $f^{-1}$. Here, *L* denotes the loss function, quantifying the difference between the estimated and actual velocity models. Our learning-based inversion approach leverages a DL (convolutional network-based) model as the function $g_{\theta }\left(\cdot \right)$. θ includes the network’s weights and biases, which are learned using a large dataset of *N* seismic data features, ${\bar{y}}_{i}$, and corresponding velocity model label, ${\bar{x}}_{i}$, pairs $\left\{\left({\bar{y}}_{i},{\bar{x}}_{i}\right),i=1,\ldots ,N\right\}$. By minimizing *L* over the entire training set, an optimal set of weights and biases, $\theta^{*}$, is obtained, at which point $g_{\theta^{*}}\left(\cdot \right)$ serves as an approximation of $f^{-1}$. Applying $g_{\theta^{*}}\left(\cdot \right)$ to the original observed signal y results in an estimate of the input space:(2)$$\begin{array}{c} x^{*}=g_{\theta^{*}}\left(y\right)\approx f^{-1}\left(y\right),\;\;\mathrm{where}\;\;\theta^{*}=\underset{\theta }{arg min}\frac{1}{N}\sum_{i=1}^{N}L\left({\bar{x}}_{i},g_{\theta }\left({\bar{y}}_{i}\right)\right), \end{array}$$
where $x^{*}$ is an approximation of the true velocity model x. This approach is illustrated in Figure 2.

### 2.3. Network Design

Our learning-based inversion approach leverages a network composed of convolutional layers, which are known for their ability to capture spatially correlated input features, such as those in seismic data. Our DL architecture is a simple encoder–decoder design with no skip connections or fully connected layers, therefore slightly departing from the widely used UNet [44] architecture, popular for machine vision tasks, but over-parameterized for our problem. The network architecture is summarized in Figure 2c. The number of trainable parameters is just over six million.

### 2.4. Feature and Label Design

The input data features are adjacent 2D seismic shot gathers. When combined, these shots form a 3D tensor, denoted as $y_{i}\in R^{n_{s}\times n_{r}\times n_{t}}$. In this expression, $n_{s}$, $n_{r}$, and $n_{t}$ represent the number of seismic shots per feature, the number of receiver measurements per shot, and the number of time measurements per shot, respectively. During the Tiber data acquisition, 3D shots were gathered using a swath of hydrophone streamers towed by the seismic vessel. For this study, we focused on a windowed 2D line of hydrophone measurements directly behind the vessel. We downsampled this line to $n_{r}=128$ measurement locations, each separated by 60 m in the x-direction. The seismic data were highcut at 20 Hz, decimated to a 24 ms sampling rate, and windowed to $n_{t}=256$ samples starting after 1.5 s of recording time. The 3D input feature, $y_{i}\in R^{64\times 128\times 256}$, is formed by $n_{s}=64$ windowed and bandpassed shot gathers, each separated by 125 meters in the x-direction. A representative input feature is illustrated in Figure 2a.

Output labels are 2D velocity models probed by their respective input data features. The 64 seismic shots, which compose one input feature, extend 17 km along the surface from the first source to the final receiver. The chosen recording time allows the seismic data to probe 12 km deep. Therefore, each output label is a 17 × 12 km velocity model slice, denoted as ${\bar{x}}_{i}\in R^{n_{x}\times n_{z}}$. Here, $n_{x}$ and $n_{z}$ represent the number of samples in the x- and z-directions, respectively. We chose $n_{x}$ and $n_{z}$ as 128 to control the resolution of the output space, making ${\bar{x}}_{i}\in R^{128\times 128}$.

### 2.5. Train, Validation, and Test Data Split

The legacy velocity model and seismic data, depicted in Figure 1, demonstrate a noticeable overlap of seismic data, and some regions of the legacy model are absent. To address this, the legacy model and seismic data are divided into two groups or folds: the training and test folds. The training fold encompasses seismic data that overlap with complete portions of the velocity model, whereas the test fold includes seismic data covering incomplete sections of the legacy model. The division process is performed spatially, taking into account inline numbers.

In seismic data collection, an inline is a set composed of seismic shot (mechanical source of medium perturbation) gathers from a single pass of the seismic vessel, where all shots align. In training datasets, the inline numbers range from 800 to 1130, comprising between 116 and 118 shot gathers each. As previously described, we group 64 shot gathers to make one input feature. Note that 64-shot features overlap with a stride of 1, yielding between 53 and 55 features per inline. Table 1 summarizes this fold division based on inline numbers, and Figure 3a offers a visual representation of the train and test fold division overlaid on the legacy velocity model.

The rationale behind this spatial division of training, validation, and test fold is to mimic the non-random nature of seismic data acquisition. Typically, seismic data are gathered in blocks, expanding onto previous surveys rather than being acquired randomly. Moreover, a random division could simplify the learning task excessively due to overlapping shot gathers within an inline. Importantly, this spatial division enables us to evaluate the network’s ability to effectively generalize to entirely new and unseen data.

## 3. Training Datasets

Several datasets are employed to train DL models, with each one’s performance evaluated on the test fold. The distinct input features define the differences between the training datasets, which are never mixed. A summary of these different training datasets can be found in Table 2. Every DL model is trained using the same 2D slices extracted from the legacy Tiber velocity model as labels.

### 3.1. Field Training Data

The training fold of the Tiber dataset (fully described in Section 2.5) was chosen such that the seismic data fully overlap with complete portions of the legacy velocity model. This overlap means that the seismic data can be associated with a velocity model label and used for training a DL model. Figure 3b illustrates how 64 adjacent shot gathers and a 2D slice of the legacy velocity model are extracted from training and validation inlines to form a labeled feature–label pair, (${\bar{y}}_{i}$,${\bar{x}}_{i}$). With an inline stride of one shot-gather, 1122 training samples and 108 validation samples form training dataset 1, outlined in Table 2. Figure 4a illustrates a representative feature from this dataset.

### 3.2. Synthetic Training Data

This study uses the acoustic and elastic wave equations to create labeled, synthetic training data to explore the feasibility of training DL models on synthetic seismic data and applying them to field seismic data. Wave equation solvers map velocity model labels to seismic shot gather via
(3)$$\begin{array}{c} {\bar{y}}_{i}=\tilde{f}\left({\bar{x}}_{i}\right), \end{array}$$
where $\tilde{f}\left(\cdot \right)$ is a finite difference approximation of the seismic experiment, using either the acoustic or elastic wave equation. By using the exact acquisition parameters of the field shot gathers, this process, illustrated in Figure 3c, yields matching sets of synthetic and field shot gathers. The synthetic gathers are combined to create input features in the same fashion as the field gathers, yielding synthetic training datasets 2–5, outlined in Table 2.

In exploration seismology, the elastic isotropic equation of motion, often used under the assumption of the subsurface behaving as an elastic isotropic medium, is as follows:(4)$$\begin{array}{c} \begin{array}{cc} \rho \left(x\right)\frac{\partial v_{i}\left(x,t\right)}{\partial t} & =\frac{\partial \sigma_{ik}\left(x,t\right)}{\partial x_{k}}+f_{i}\left(x,t\right),\\ \frac{\partial \sigma_{ij}\left(x,t\right)}{\partial t} & =\lambda \left(x\right)\frac{\partial v_{k}\left(x,t\right)}{\partial x_{k}}\delta_{ij}+\mu \left(x\right)\left[\frac{\partial v_{i}\left(x,t\right)}{\partial x_{j}}+\frac{\partial v_{j}\left(x,t\right)}{\partial x_{i}}\right]+m_{ij}\left(x,t\right), \end{array} \end{array}$$
where the Einstein summation convention is used. In this equation, x and *t* represent spatial and temporal coordinates, respectively, while $\delta_{ij}$ denotes the Kronecker delta. Three elastic parameters are employed to fully characterize the subsurface: density (ρ), first Lamé parameter (λ), and shear modulus (μ). The wavefield variables include the particle velocities ($v_{i}$) and the symmetric stress tensor components ($\sigma_{ij}$), with $\sigma_{ij}=\sigma_{ji}$. The wave propagation arises from the source terms $f_{i}$ and $m_{ij}$, which symbolize a volumetric force field and the moment tensor’s time derivative, respectively [45]. Hydrophone pressure data can modeled by sampling the diagonal stress tensor components of the elastic wavefield.

Another common assumption in seismic exploration is that the Earth behaves as an acoustic medium with constant density, which simplifies Equation (Equation 4) to the acoustic isotropic constant density wave equation:(5)$$\frac{1}{v^{2}\left(x\right)}\frac{\partial^{2}p\left(x,t\right)}{\partial t^{2}}-\nabla^{2}p\left(x,t\right)=s\left(x,t\right),$$
where *v* is the seismic wave velocity, *p* is the wavefield parameter (pressure), and *s* is the acoustic source function. These wave equations present trade-offs. By making fewer assumptions about wave physics, the elastic wave equation enables the generation of more realistic-looking synthetic data. However, the calculation is computationally demanding and requires we make assumptions elsewhere, namely, regarding the elastic earth parameters (ρ, λ, and μ) since the legacy model is only parameterized by acoustic velocity. Here, we use Gardner’s relation to derive a density model and we assume a constant ratio between shear and acoustic wave velocities [46]. Conversely, while more computationally efficient, the acoustic wave equation simplifies seismic wavefield modeling by forgoing elastic effects, which may result in less accurate reflections of complex geological structures. Figure 4b–e illustrates example synthetic input features from training datasets 2 and 4, which use the acoustic wave equation, and training datasets 3 and 5, which use the elastic wave equation.

### 3.3. Synthetic Training Data with Improved Reflection Events

As seen in Figure 4a, a dominant trait of the field data features are the reflections from sediment layers situated above the salt, within the salt basin, and beneath the salt. However, the legacy Tiber velocity model lacks high wavenumber features, such as sediment reflectors, causing the synthetic data derived from the legacy model to miss these crucial reflection events.

We employed a geologic model builder to enhance the legacy model by simulating the deposition of sedimentary layers [47]. This builder mimics sedimentary deposition by selecting seismic wave velocity, density, layer thicknesses, and layer folding from user-defined distributions, generating realistic yet random sediment models. While no single random model is expected to match reality, our approach aims to create a range of sediment models that spans the distribution of possible real-world sediment models in the Tiber region. Furthermore, since we use many sediment models for feature generation, we mitigate overfitting to any specific deposition model. We chose the distributions for the rock and layer properties to ensure that the modeled seismic data closely resembled the real data in amplitude and visual features, as illustrated in Figure 4. Figure 4d,e display example synthetic input features from training datasets 4 and 5, which incorporate these enhanced reflection events.

### 3.4. Data Augmentations

The implementation of data augmentation is critical in training DL models, enhancing their generalization ability and mitigating overfitting. Our approach utilizes three primary training data augmentations techniques: horizontal flipping, random bandpassing, and two-dimensional wave-propagation correction. Different combinations of these augmentations are applied to the synthetic training datasets 2–5.

Horizontal flipping serves to counteract potential model bias due to the seismic vessel’s sailing direction, enhancing robust learning. Meanwhile, random bandpassing helps prevent the neural network from overfitting to the specific wavelet used to generate synthetic training samples by introducing variability in the wavelet shapes. With this augmentation, each feature receives a random lowcut between 3 and 9 Hz and a random highcut between 14 and 20 Hz before being presented to a DL model. Two-dimensional wave-propagation corrections are employed to address the phase, spectrum, and amplitude effects caused by simulating wave propagation in only two dimensions [48].

Furthermore, we apply an augmentation to the test fold data, noise reduction via structural smoothing, to further bridge the gap between synthetic training data and field test data [49]. As the name suggests, this method smoothes along coherent energy or structure, removing both coherent and incoherent noise in the test fold data. This can make the field test data appear more synthetic.

## 4. Performance Evaluation

### 4.1. Training Policy and Experimental Setup

We trained various DL models, all with the architecture illustrated in Figure 2 and the training parameters in Table 3, using the training datasets outlined in Table 2. When synthetic training datasets are used, we used various combinations of data augmentations during training and testing. We performed a qualitative, quantitative, and geophysical evaluation of velocity model predictions made in the test fold to evaluate a given network’s performance. Quantitative metrics can be calculated for samples in the test fold with corresponding labels from the legacy velocity model. However, since the legacy velocity model is incomplete in the test fold, some test fold features are without labels, making quantitative comparisons impossible or incomplete. Therefore, we also relied on a qualitative interpretation of the geologic feasibility of test fold velocity model predictions. Finally, we used a migration algorithm with the predicted velocity models to form a geophysical comparison based on the final image quality.

Training time varies slightly depending on the training time data augmentations since this adds the computational cost of altering the training data every time they are passed through the network. In general, it takes approximately 75 s per epoch, totaling around 100 min to train one network. This time also varies based on the batch size and the number and the model of GPUs used for training. The batch size is limited by GPU memory. With 3D features and the deep convolutional network in Figure 2, memory usage per GPU approaches the the 32 GB limit of the available NVIDIA V100s. More modern GPUs with larger memory limits would not have this issue and would allow for faster training time.

### 4.2. Quantitative Comparison

We used three key metrics for evaluating the performance of our DL models: Mean Squared Error (MSE), Structural Similarity Index Measure (SSIM), and $R^{2}$ (R-squared). The MSE calculates the average of the squares of the differences between the predicted and the legacy velocity models. The SSIM metric provides a perception-based measure that considers changes in structural information [50]. Finally, $R^{2}$, known as the coefficient of determination, quantifies the goodness of fit between the predicted and the legacy velocity models. These metrics are outlined in Table 4 and plotted in Figure 5, where we specifically highlight the highest-performing models from each of the five training datasets.

### 4.3. Qualitative Comparison

Considering that a quantitative analysis is only feasible where predictions overlap with the incomplete legacy model, a qualitative comparison was also performed. This process involved visually comparing the velocity predictions with the legacy model and assessing the geological feasibility of the predictions in areas where the legacy model is unavailable. Figure 6 showcases the results for three standout DL models from training datasets 1, 4, and 5, whose quantitative results are highlighted in bold in Table 4. For each standout DL model, one test sample from four different inlines is plotted, with inlines varying in distance from the training set. Spatial distance from the training dataset directly influences the model’s ability to generalize.

### 4.4. Geophysical Comparison

For the geophysical comparison, we selected two of the highest-performing models: one trained on dataset 1 and the other on dataset 5. Using an ensemble regression approach, we aggregated the predictions from complete inlines [51]. We then employed a widely used seismic imaging method, Reverse Time Migration (RTM), using the ensembled velocity models as inputs [52]. Due to the 2D nature of the DL model outputs, our RTM analysis is restricted to two dimensions. Figure 7 illustrates the migration results from the two DL models for two inlines, one near the training set and one far from it. The migrations are superimposed on the ensembled velocity models used for the migration algorithm, demonstrating how migrated reflections align with the velocity models, which is a partial measure of the success of this analysis method.

## 5. Results

We consider the performance of convolutional DL models trained with synthetic and field data using quantitative, qualitative, and geophysical comparison methods.

The results of the quantitative comparison are best analyzed through the scatter plot of test fold MSE and SSIM scores in Figure 5. The selection of metrics is not aleatory, as MSE represents overall fit and SSIM structural reconstruction quality. Several key insights can be derived from the quantitative comparison. First, the DL model trained on field data clearly outperforms the others, not only in MSE and SSIM, but also with the highest $R^{2}$ score of 0.909. This $R^{2}$ score underscores the excellent fit of this model to the legacy velocity model. Second, when all other factors are held constant, models trained on synthetic data perform better when the data ertr created using the elastic wave equation compared to those using the acoustic wave equation. The addition of enhanced sediment geological priors in the synthetic training data has led to better performance across all three metrics, including $R^{2}$. Finally, the role of data augmentation techniques is nuanced; while they sometimes lead to inconsistent effects on model performance, they were required to produce the highest performing models trained on synthetic data.

The qualitative comparisons, made for three standout DL models in Figure 6, reaffirm that the DL model trained with field data demonstrates predictions most akin to the available segments of the legacy velocity model. Moreover, both field and elastic dataset-trained DL models yield geologically plausible predictions where the legacy model lacks data for comparison. The acoustic dataset-trained DL model, although providing reasonable predictions, overlooks an evident salt canyon, suggesting the impact of the domain gap between synthetic training data and field test data.

Finally, the RTM imaging result in Figure 7 indicates that the velocity models created from both the DL model trained with field data and the DL model trained with synthetic data are sufficient to create an initial seismic image of the subsurface. The migration results from seismic inline 1040, Figure 7a,b, have coherent reflections above the salt, within the salt basin, and beneath the salt. These reflections are present, but become less coherent, in Figure 7c,d, which illustrates results from seismic inline 1120. Inline 1040 is spatially closer to the training fold than inline 1120, implying that the spatial distance between training and testing data is related to ability of a DL model to generalize to unseen data.

## 6. Discussion

Our work provides a comprehensive analysis of the performance of DL models trained using both synthetic and field data. The quantitative, qualitative, and geophysical results convincingly demonstrate that models trained on labeled field data, dataset 1, perform superiorly across various metrics, including MSE, SSIM, and $R^{2}$. These models present a geologically coherent match with legacy models, leading to migration images with high coherence.

On the other hand, models trained exclusively on synthetic data, datasets 2–5, demonstrate plausible and consistent velocity model inference results, even though they do not quite reach the benchmark set by models trained with field data. The synthetic training data-based models generate migration images that facilitate the initial interpretation of salt structures and sedimentary basins within the salt. These outcomes underscore the value of synthetic data, but also point to the need for ensuring that the domain gap between the synthetic training data and field test data is appropriately addressed. We accomplished this by introducing reflection events to the synthetic training data, substantially improving the model’s performance.

Our results suggest that the key attributes of the feature space must be comprehensively understood and incorporated when generating synthetic training data. The use of wave equation solvers that more closely resemble real-world physics can dramatically improve the accuracy of synthetic data. In our case, using the elastic wave equation over the acoustic wave equation resulted in more accurate velocity model predictions from a qualitative and quantitative perspective.

The implications of our findings are significant, particularly for geoscience companies that possess extensive labeled seismic data. These entities have invested substantial resources in data acquisition and the construction of high-fidelity earth models. The ability to repurpose these data to train machine learning models that can generalize across different geological regimes and seismic acquisition parameters could fundamentally accelerate the inversion of velocity models from seismic data. This acceleration could lead to substantial cost reductions in subsurface projects, making a notable impact on emerging fields such as enhanced geothermal systems, geologic carbon sequestration, and renewable gas storage.

For organizations with limited access to labeled seismic data, our study offers a viable alternative—the use of synthetic seismic data to train machine learning models. As we have shown, the performance of models trained on synthetic data can approach that of models trained on field data, especially if the wave equation solvers used in creating synthetic data incorporate a comprehensive set of earth physics parameters, including anisotropy and attenuation. In some cases, synthetic data might even be the preferred training dataset since the velocity model labels associated with field data cannot be entirely validated in all subsurface regions.

Looking forward, several areas of future work emerge from our study. Feature engineering could potentially further close the domain gap between field and synthetic data, enhancing the performance of models trained on synthetic data. Another valuable extension would be to engineer solutions that can handle three-dimensional seismic data. Lastly, there is a need to evaluate the performance of synthetic training data created with more advanced wave equations, which could further enhance the performance of models trained on synthetic data.

## 7. Conclusions

In this study, we investigated the efficacy of DL models in seismic velocity model-building using both synthetic and real-world field data. Our results revealed that models trained with field data consistently outperformed those trained exclusively with synthetic data. However, synthetic data still proved valuable, particularly when the feature space was carefully crafted and more realistic wave equations were used. This research showed the feasibility of using machine learning to recover accurate seismic velocity model profiles from field-recorded seismograms in a field application with complex geology. Additionally, our findings highlighted the potential of reusing existing seismic data for training machine learning models. This approach could offer advantages in terms of cost and time-efficiency in subsurface reservoir imaging. Future work should consider refining feature engineering for synthetic data, extending the methodologies to three-dimensional seismic data, and investigating advanced wave equations for enhanced synthetic data generation.

## Figures and Tables

**Figure 1 sensors-23-08277-f001:**
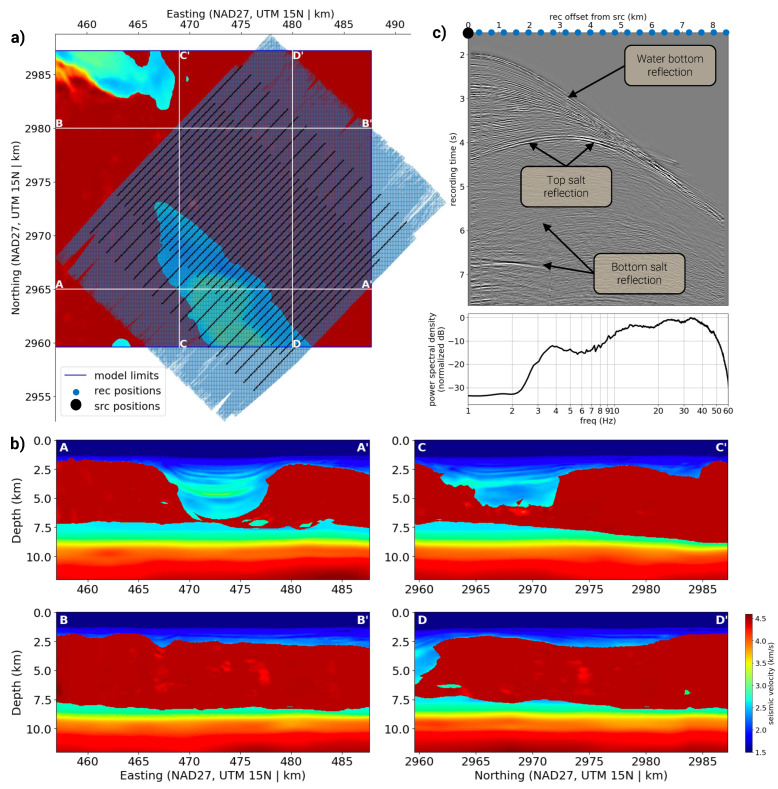
An overview of the legacy velocity model and seismic recordings in the Tiber dataset. Panel (**a**) depicts a depth slice of the legacy velocity model at a depth of 4.5 km, displaying an extensive allochthonous salt body (indicated in red, with velocities exceeding 4.5 km/s). Velocity cross-sections, A-A’, B-B’, C-C’, and D-D’, in Panel (**b**) expose the thickness of the salt body, reaching over 5 km in some areas and largely blocking the illumination of crucial reservoir compartments beneath. The unique positions of all seismic sources and receivers are portrayed in Panel (**a**), indicating that the data overlap only partially with the given velocity model. Panel (**c**) showcases a representative 2D line of hydrophone recordings captured using hydrophone pressure sensors and airgun sources, and the power spectrum density averaged over all recordings.

**Figure 2 sensors-23-08277-f002:**
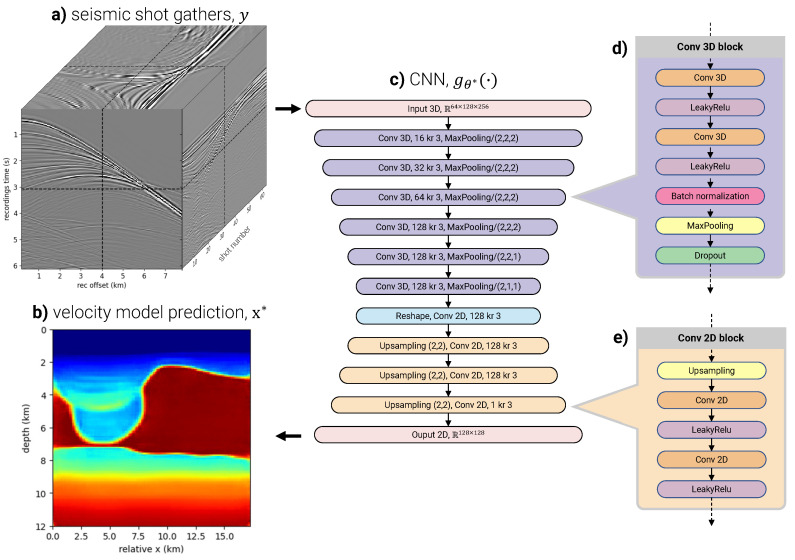
Overview of the learning-based seismic velocity inversion method. (**a**) The input feature space comprises a set of 2D seismic shot gathers which form a 3D tensor, $y_{i}\in R^{64\times 128\times 256}$. (**b**) The output is a 2D seismic velocity model, denoted as $x_{i}^{*}\in R^{128\times 128}$, which corresponds to the area probed by the input shot gathers. (**c**) The DL architecture is purely convolutional and composed of an encoder (six top layers) featuring blocks of 3D convolutional layers (**d**), and a decoder composed of 2D convolutional blocks (**e**).

**Figure 3 sensors-23-08277-f003:**
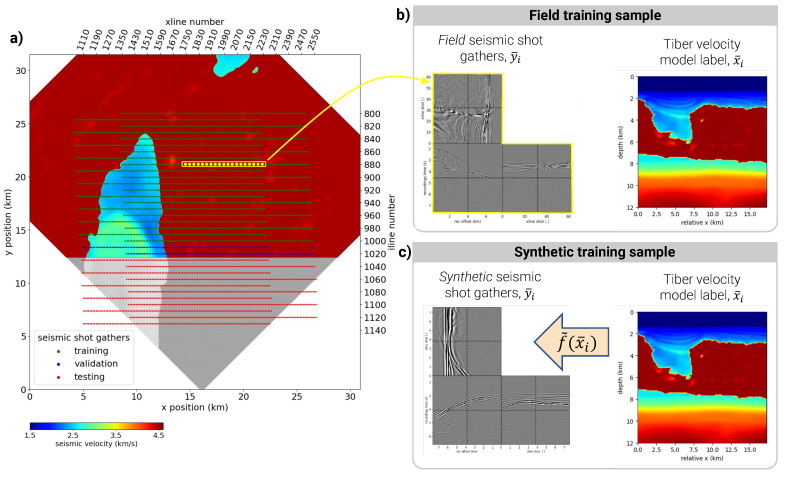
(**a**) A depth slice of the legacy velocity model at 4.5 km and the seismic data source locations, rotated to align the seismic data inlines with the x-axis. The color-coding of each source location signifies its classification into one of the learning folds: training, validation, or test. Folds are divided according to the seismic inline, with training and validation folds selected where seismic data and complete sections of the legacy velocity model overlap. The test fold includes inlines overlapping with the incomplete portion of the legacy velocity model, indicated in grayscale. Further details about the fold division can be found in Table 1. (**b**,**c**) Feature/label pairs from two different training datasets; (**b**) uses field recorded seismic data as input features and **(c)** uses synthetic seismic data as input features, modeled with a finite difference approximation of the seismic experiment, $\tilde{f}\left(\cdot \right)$.

**Figure 4 sensors-23-08277-f004:**
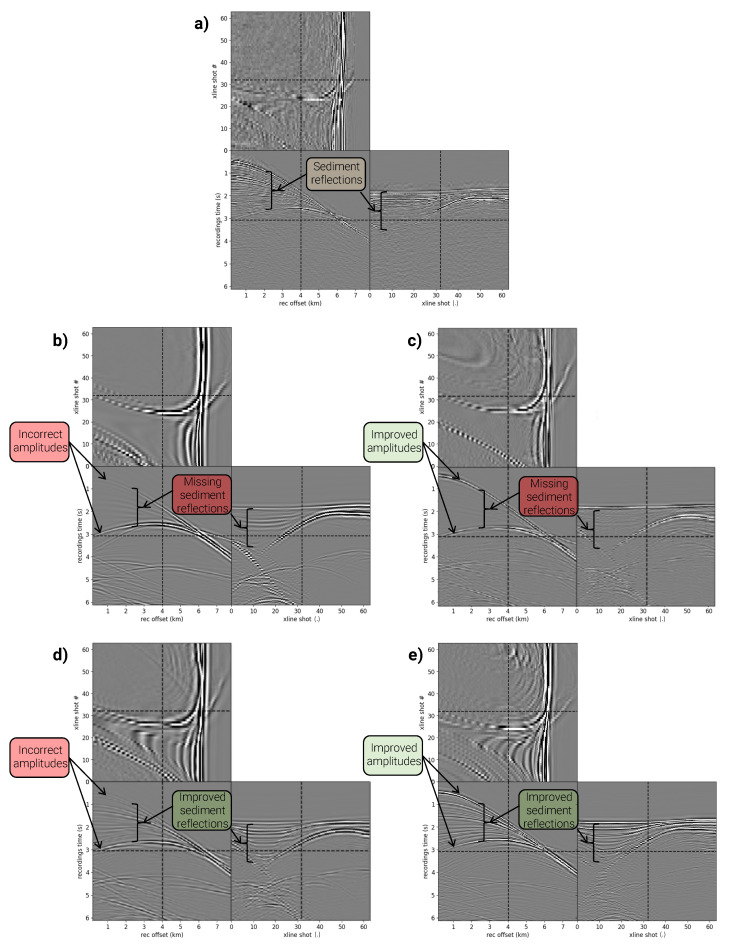
Matching features from training datasets 1–5 in order through panels (**a**–**e**). Synthetic data created using the elastic wave equation (**c**,**e**) provide more accurate arrival amplitudes, closely mirroring the field data (**a**), thanks to the improved physics modeling of the elastic wave equation. Synthetic data incorporating improved sedimentary priors (**d**,**e**) capture essential reflection events present in the field data (**a**), which are absent in the synthetic versions in panels (**b**,**c**). Enhancing the realism of synthetic data is paramount for establishing a synthetic training dataset that effectively trains a DL model for application to real-world field data, thus bridging the domain gap between synthetic training and field application.

**Figure 5 sensors-23-08277-f005:**
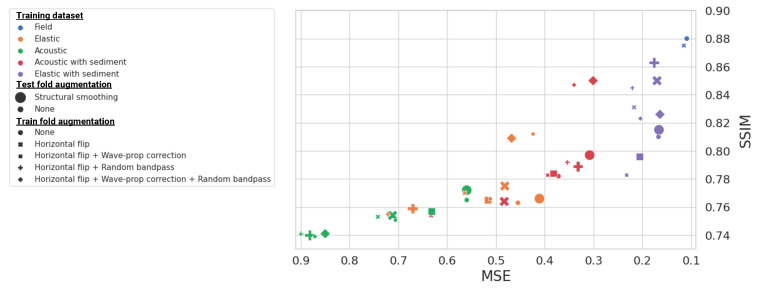
A scatter plot of the test set quantitative scores for all trained DL models from Table 4. Each point corresponds to a different DL model, with the x-axis representing MSE while the y-axis represents SSIM metric. A lower MSE is better, but the opposite situation occurs for SSIM; therefore, the sweep spot is located on the right-most column.

**Figure 6 sensors-23-08277-f006:**
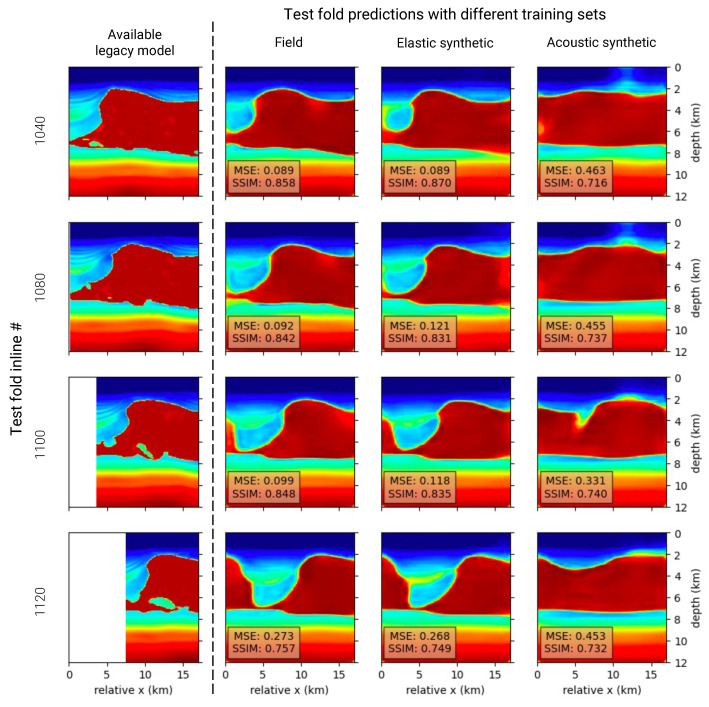
Qualitative comparisons of velocity model predictions from the test fold, with each row representing a different seismic inline sample from the test fold. The first column depicts the legacy velocity model, functioning as the ground truth for each sample. Due to the incompleteness of the legacy velocity model, parts of the samples are unavailable. The subsequent three columns represent predictions from three distinct DL models, each excelling in their individual categories. These categories include models trained on field data, synthetic data employing the acoustic wave equation, and synthetic data using the elastic wave equation. Each prediction is accompanied by its respective MSE and SSIM scores.

**Figure 7 sensors-23-08277-f007:**
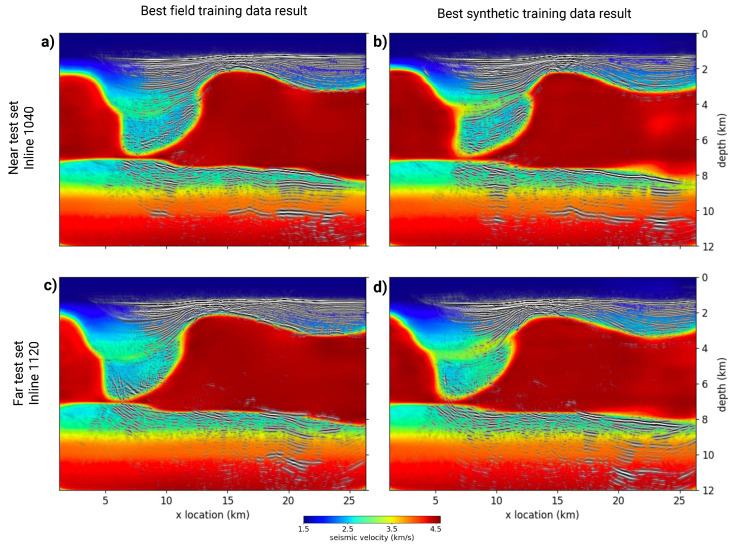
Geophysical imaging comparison of velocity model predictions from two different seismic inlines within the test fold—one spatially proximate, Panels (**a**) and (**b**), and one distant, Panels (**c**) and (**d**), to the training data. The two columns represent predictions from standout DL models from datasets 1 and 5. Each image results from the ensembling of an entire inline of velocity model predictions into a 2D prediction spanning the entire x-axis. These ensembled predictions served as the migration velocity model for seismic reverse time migration of the respective field data from the inlines. Overlaying the migration image onto the ensembled migration velocity model reveals how migrated reflection events correspond with their respective velocity model predictions.

**Table 1 sensors-23-08277-t001:** The division of shot gather inlines within the training, validation, and test folds. The test fold was chosen where shot gathers overlap incomplete portions of the legacy Tiber velocity model. The remaining inlines are placed in the training fold.

Fold	Inline Range	Number of Seismic Shots	Number of Feature–Label Pairs
Train	800–1000	2679	1122
Validation	1010–1020	234	108
Test	1030–1130	1285	592

**Table 2 sensors-23-08277-t002:** High level descriptions of the different training datasets used in this study. The features vary between datasets, but the labels remain the same.

ID Number	Training Data	Feature Space Description
1	Field	Field gathers from the Tiber dataset
2	Acoustic	Synthetic gathers made with the acoustic wave equation
3	Elastic	Synthetic gathers made with the elastic wave equation
4	Acoustic with sediment reflections	Synthetic gathers made with the acoustic wave equation and improved sediment priors
5	Elastic with sediment reflections	Synthetic gathers made with the elastic wave equation and improved sediment priors

**Table 3 sensors-23-08277-t003:** Parameters used to optimize the weights of the DL model summarized in Figure 2. These parameters are held the same for all training experiments, while training datasets and data augmentations vary.

Training Parameter	Value
Optimization algorithm	Adam
Number of epochs	80
Loss function	MSE
Learning rate	0.00005
Batch size	24
GPUs	4 × NVIDIA V100s

**Table 4 sensors-23-08277-t004:** Test set quantitative scores for DL models trained using different training datasets, training feature augmentations, and test feature augmentations. The weighted average MSE (grey background) and SSIM (white background) scores are calculated by comparing DL velocity predictions with the available portions of the Tiber legacy model in the test fold. However, as these quantitative metrics can only be calculated where the incomplete legacy model overlaps with the test set predictions, evaluating model performance should also incorporate qualitative and geophysical assessments. This can be achieved by visually comparing velocity predictions and the coherence of resulting migrated images. One standout DL model from each training dataset is highlighted in bold. While these models might not have the best MSE and SSIM metrics in their respective categories, they were chosen because they excel across all three performance indicators: quantitative measures (MSE, SSIM, and $R^{2}$), qualitative assessment (geological feasibility), and geophysical evaluation (migration image coherency). The $R^{2}$ scores are calculated for these five DL models, which are 0.909, 0.558, 0.662, 0.761, and 0.860, respectively.

Data Augmentation		Training Dataset				
Test Fold Augmentation	Train Fold Augmentation	Field	Acoustic	Elastic	Acoustic with Sediment	Elastic with Sediment
None	None	0.109	0.560	0.455	0.372	0.167
		0.880	0.765	0.763	0.782	0.810
	Horizontal flip	**0.115**	0.742	0.564	0.633	0.217
		**0.875**	0.753	0.770	0.754	0.831
	Horizontal flip + Wave-prop correction		0.707	0.512	0.395	0.233
			0.751	0.766	0.783	0.783
	Horizontal flip + Random bandpass		0.900	0.722	0.354	0.221
			0.741	0.755	0.792	0.845
	Horizontal flip + Wave-prop correction + Random bandpass		0.871	**0.424**	0.340	0.204
			0.739	**0.812**	0.847	0.823
Structural smoothing	None		**0.560**	0.411	0.308	0.166
			**0.772**	0.766	0.797	0.815
	Horizontal flip		0.712	0.482	0.483	0.170
			0.754	0.775	0.764	0.850
	Horizontal flip + Wave-prop correction		0.632	0.517	0.382	0.206
			0.757	0.765	0.784	0.796
	Horizontal flip + Random bandpass		0.882	0.671	0.332	**0.176**
			0.740	0.759	0.789	**0.863**
	Horizontal flip + Wave-prop correction + Random bandpass		0.850	0.468	**0.301**	0.164
			0.741	0.809	**0.850**	0.826

## Data Availability

The seismic data and results are made available at and are reproducible from the following git repository: https://premonition.stanford.edu/sfarris/vmb-net (accessed on 1 October 2023).

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
