# Peer review of "Learning-Based Seismic Velocity Inversion with Synthetic and Field Data"

_sensors, 2023, doi:10.3390/s23198277_

Round 1
Reviewer 1 Report
The manuscript shares the experiences the authors had in using ML to invert for velocity models directly from (complex) field data. The paper is generally well written and insightful. My major concern is with regard to the claims of uniqueness/novelty of their tests and application on real data. I strongly encourage the authors to acknowledge other investigations along these lines, some of which I share in the annotated version for paper, and be slightly conservative in their claims, especially those related to the novelty of their investigations. The work provides a lot of valuable insights related to the way the authors investigated this topic, which is definitely important, and it does not require exaugurated statements on Novelty, like those in the “novel contributions” and “conclusions” sections. In addition, a fair review of other’s work will only strengthen the messages included in this paper.
Another concern involves the illumination region of the data. Its seems that the limited range of data used for the input features do not cover the much larger extent of the velocity model labels forcing the network to learn fixed behavior of the models along the sides that are not related to data features. This is a form of prior that the authors need to highlight, and thus, the predicted extensions of the model come from that component of the learning, not from the data. This can be seen in Figure 7, where obviously, the migration image does not cover big parts of the left side of the model, and yet, we end up with interesting model changes there.
Though the paper is well written, the abstract and conclusions need a rewrite. For me, it is important that the authors personalize the abstract with words like “We”. The abstract often starts with a sentence or two on motivation, and then describes the development and a summary of results. Statements like “In the conclusion ….” is not abstract language. In fact, the abstract is where we include the most important conclusions naturally, without describing what we will have in the conclusion. On the conclusion side, it should summarize key findings and results, not boost about the potential impact of the work. That is left to the reader.
In the some of the Figure captions (like Figures 5 and 6), the authors draw conclusions from what they see in the Figure, and usually this is reserved for the text. Figure captions are used to only describe the Figure and clarify it.
I include additional corrections and suggestions in the uploaded annotated in which I encourage the authors to address them, and make the necessary revision to the paper.

Author Response
Thank you for your constructive feedback. We appreciate the effort you've invested in reviewing our manuscript. Below, please find our responses to your comments and suggestions:
On the Novelty of the Investigations
We are grateful for your suggestions about acknowledging related work and being more conservative in our claims of novelty. In response, we have included the additional recommended citations that are directly relevant to our work. Additionally, we've adjusted our discussion of the study's novelty. While we don't claim to be the first to use machine learning with field data for velocity model inversion, we believe our contribution lies in doing so in a complex geologic regime with salt structures.
On the Illumination Region of the Data
You make a valid point regarding the illumination issue seen in Figure 7. The RTM image indeed has a higher fold near the middle of the inlines, and when overlaying the image over the velocity model, low amplitudes had to be clipped. Your observation about the reduced quality near the edges is accurate; however, we can see that the edges of the salt structures are still illuminated in the zero-offset shot gathers, as evidenced by a figures from my thesis which I include in my full response but not in the manuscript.
On the Abstract and Conclusions
Based on your suggestions, we've significantly reworked the abstract and conclusion sections. Your insights have been invaluable in improving the quality and clarity of both.
On Figure Captions
We have taken your advice and moved our interpretations from the figure captions into a newly created Results section. In general, I like figures to stand on their own, able to be mostly understood without reading the text. I therefore tend to make captions verbose and included too much information in this case. But, you are correct that interpretations should be reserved for the Results section.
Thank you once again for your valuable feedback, which has undoubtedly strengthened our manuscript.

Reviewer 2 Report
This manuscript is well written and is interesting for geophysists engaged in velocity construction. It is acceptable except for some minor points should be revised.
1. Move Tabel s' captions to the position above the tables.
2. Delete the last ";" at the end of Keywords.
3. Give the full words of the abbrebiation of CNN when it is first used.
Author Response
Thank you for your feedback. We responded to your comments in the attached pdf.

Reviewer 3 Report
The manuscript showcases deep learning based seismic velocity inversion in a case study style on a specific field data. It provides a comprehensive analysis of CNN on seismic inversion, such as field data, synthetic data, data augmentation et al. The results add value for researchers on this direction. I have following comments:
C01: In the abstract, the statement “Methods involve training various DL models with different datasets” is not precise and misleading. Please point out that the various DL models are in same structure and training scenarios.
C02: In Table 2, please make detail explanation on training data with/without sediment reflections. Is it in the velocity profile? If so, velocity profile with/without sediment reflections should be demonstrated. And how to get those sediment reflections?
C03: Where are the parameters coming from, density, Lame parameter and shear modulus for the elastic equation? What is the relationship between our inversion target P wave velocity? The wavefield of the elastic wave equation is velocity, it is vector and then how to match with the field seismic data? There is only P wavefield for the field data. The model trained with synthetic data with elastic wave equation only use one component of the elastic wavefield?
C04: And for the augmentation to the test fold data, is this strategy also for the field data test? Example of this augmentation should be demonstrated since this is very important to mitigate data discrepancy.
C05: Please add details on the random bandpass augmentation operation.
C06: For the section 4.4 geophysical comparison, there is no comments on the image quality of the migration profile using different velocity. It seems there is good and bad portions for both images. Please give comments. And for the RTM, it is 2D or 3D migration?
C07: Since the sediment reflections are added in the training data, then the inversion results should include these sediment structures in the model. There is no mention on that in the numerical test part. Please give comments on this.
Author Response
Thank you for your thorough and constructive feedback. We truly appreciate the time and effort you've invested in reviewing our manuscript. Please see the attached pdf where we reply to your comments.

Reviewer 4 Report
sensors-2601995 REVIEW
This paper is a thorough study that examines the application of deep learning to the construction of velocity models trained on field data (as well as synthetic data) from geological environments that are more complex (reservoirs overlain by complex geology) than those used in previous work. The study extends the usefulness of deep learning in seismic inversion. The paper is very well written and organized and is basically written for those familiar with deep learning theory and techniques. The paper can be published in its present form, or perhaps after some very minor corrections (see my minor comments below).
COMMENTS:
[1] Lines 38-39: For those not very familiar with deep learning, it would be useful to define or describe in a bit more detail terms like “labelled training dataset” and “output labels”.
[2] Figure 3: Descriptions of Figures 3b and 3c appear to be missing from the caption of Figure 3.
[3] Line 132: “seismic data are gathered” ? (“data” is plural). See also lines 39, 60, 119, 145, 286 and perhaps other places.
[4] Line 156: Technically, it’s called the elastic isotropic equation of motion.
Author Response
Thank you for your constructive feedback. We appreciate the effort you've invested in reviewing our manuscript. Please see the attached pdf where we reply to your comments.

Round 2
Reviewer 1 Report
The authors have generally addressed my concerns and made the necessary revision. I have only very minor typo corrections in the annotated version of their paper, which the authors can include as part of submitting their paper for publications.
